# Genomic Consequences of Fragmentation in the Endangered Fennoscandian Arctic Fox (*Vulpes lagopus*)

**DOI:** 10.3390/genes13112124

**Published:** 2022-11-15

**Authors:** Christopher A. Cockerill, Malin Hasselgren, Nicolas Dussex, Love Dalén, Johanna von Seth, Anders Angerbjörn, Johan F. Wallén, Arild Landa, Nina E. Eide, Øystein Flagstad, Dorothee Ehrich, Aleksandr Sokolov, Natalya Sokolova, Karin Norén

**Affiliations:** 1Department of Zoology, Stockholm University, 10691 Stockholm, Sweden; 2Centre for Palaeogenetics, Svante Arrhenius väg 20C, 10691 Stockholm, Sweden; 3Department of Bioinformatics and Genetics, Swedish Museum of Natural History, 11418 Stockholm, Sweden; 4Norwegian Institute for Nature Research, 7485 Trondheim, Norway; 5Department of Arctic and Marine Biology, UiT Arctic University of Tromsø, 9037 Tromsø, Norway; 6Arctic Research Station of Institute of Plant and Animal Ecology, Ural Branch, Russian Academy of Sciences, Zelenaya Gorka Str. 21, 629400 Labytnangi, Russia

**Keywords:** inbreeding, runs of homozygosity, bottleneck, fragmentation, mutational load, conservation

## Abstract

Accelerating climate change is causing severe habitat fragmentation in the Arctic, threatening the persistence of many cold-adapted species. The Scandinavian arctic fox (*Vulpes lagopus*) is highly fragmented, with a once continuous, circumpolar distribution, it struggled to recover from a demographic bottleneck in the late 19th century. The future persistence of the entire Scandinavian population is highly dependent on the northernmost Fennoscandian subpopulations (Scandinavia and the Kola Peninsula), to provide a link to the viable Siberian population. By analyzing 43 arctic fox genomes, we quantified genomic variation and inbreeding in these populations. Signatures of genome erosion increased from Siberia to northern Sweden indicating a stepping-stone model of connectivity. In northern Fennoscandia, runs of homozygosity (ROH) were on average ~1.47-fold longer than ROH found in Siberia, stretching almost entire scaffolds. Moreover, consistent with recent inbreeding, northern Fennoscandia harbored more homozygous deleterious mutations, whereas Siberia had more in heterozygous state. This study underlines the value of documenting genome erosion following population fragmentation to identify areas requiring conservation priority. With the increasing fragmentation and isolation of Arctic habitats due to global warming, understanding the genomic and demographic consequences is vital for maintaining evolutionary potential and preventing local extinctions.

## 1. Introduction

Arctic species are at an increasing risk of extinction due to climate change and human activities [1,2]. Habitat fragmentation is a widespread consequence of global warming [3], particularly due to an expansion of the treeline and increased boreal influence in the southern Arctic [4,5]. This fragmentation is exacerbated by direct and indirect human activities, through increased development and encroachment within the Arctic itself, and industrial processes contributing to the warming climate [2]. Consequently, specialist species endemic to the Arctic undergo population fragmentation, threatening their persistence when they are confined to isolated pockets of habitat [6,7].

Connectivity is vital for dispersal and gene flow. When arctic species are confined to fragmented populations, they tend to lose genetic variation and the ability to withstand environmental and demographic stochasticity [8]. Genetic drift, inbreeding and inbreeding depression (i.e., genome erosion) are major consequences of small population size, which are likely to occur following subpopulation isolation [9]. Furthermore, Allee effects, (i.e., the effects of reduced population density on reproduction and population growth), will also arise [10]. These genetic and demographic processes affect small populations disproportionately, in what is referred to as ‘the small population paradigm’ [11]. The interplay between these processes can create a feedback loop known as ‘the extinction vortex’, where the negative impact of genetic and demographic factors will increase with decreasing population size [12,13,14,15]. Under such a scenario, genetic differentiation between subpopulations will increase, genetic variation will decrease, and both will eventually contribute to population decline and extinction through reduced fitness [14]. The most widely accepted explanation for inbreeding depression is the dominance hypothesis, i.e., the increase in homozygosity at loci with an accumulation of rare recessive deleterious alleles [16,17,18]. Alternatively, the overdominance hypothesis predicts that the inbreeding-induced loss of heterozygosity reduces fitness based on the heterozygote advantage theory [9,19]. These processes cause a reduction in average fitness of the population, which can persist for many generations.

Gene flow from a separate, differentiated source can counteract genetic drift, and mitigate inbreeding depression through increased fitness and population growth (genetic rescue [20,21]). Maintaining metapopulation structure and regular gene flow is thus vital for population persistence due to the transient fitness benefits of infrequent gene flow. Inbreeding depression and genetic rescue are challenging to document in the wild [22,23], due to the extensive monitoring required, combined with unpredictable natural conditions, and the difficulty in obtaining individual fitness data. However, some studies have documented inbreeding depression (i.e., [24,25,26,27]), and genetic rescue (i.e., [28,29]) in wild populations. Inbreeding depression is traditionally estimated using fitness-related traits linked to pedigree-based inbreeding coefficients [9,22]. However, pedigrees have many caveats, including unknown relatedness among founders and limited historical perspectives about inbreeding [30]. Thus, pedigrees are often unreliable, with variable capacity to detect inbreeding depression among studies. Advances in whole genome sequencing have provided the opportunity to assess genome-wide heterozygosity and genomic inbreeding levels. Stretches of identical by descent (IBD) chromosomal regions are characteristic of inbreeding due to recent common ancestors. These runs of homozygosity (ROH) are shortened through recombination over generations [31,32] where the shorter the ROH, the older the inbreeding, however very short ROH have a higher likelihood of being identical by state [33]. Signatures of inbreeding along the entire chromosome have recently been identified in a range of species [30,34,35,36].

Genomic analyses can also identify accumulation of putatively deleterious mutations across the genome. This ability to assess the proportion of homozygous and heterozygous genotypes of mutations in small, isolated populations relative to an outbred population is valuable for exploring the potential consequences of population fragmentation [34]. It is still debated whether inbreeding depression is caused by a few strongly deleterious mutations or many weakly deleterious mutations [30]. Nevertheless, identifying the proportion of deleterious mutations after a population bottleneck or re-colonization event with few founders can be used as a proxy for genetic load and is an important first step to understanding the underlying mechanisms of inbreeding depression.

The population of arctic fox (*V. lagopus*) in Fennoscandia (Sweden, Norway, Finland and on the Kola Peninsula) is at present highly fragmented, but was once connected to the circumpolar distribution of the species [37]. The population was drastically reduced by hunting pressure in the late 19th century [38], leading to a severe demographic and genetic bottleneck [39,40]. The increased irregularity of natural prey cycles, combined with interspecific competition and intraguild predation from red fox (*V. vulpes*) has prevented the population from recovering [41,42]. The few remaining Fennoscandian subpopulations are isolated, genetically distinct units that exhibit decreasing levels of genetic variation with increasing distance to the once connected, likely panmictic populations in Siberia [43,44].

Arctic foxes are long-distance dispersers, capable of traveling thousands of kilometers in the absence of barriers [45]. However, the Scandinavian tundra (mostly mountain tundra) is naturally fragmented with intersectional boreal forests [46,47]. Ice-free coasts around Fennoscandia and an increasingly ice-free White Sea limit dispersal from the East [48,49]. The Russian Arctic stretching from the Kanin Peninsula to the shores of the Bering Sea does not exhibit such barriers [48].

The northernmost subpopulations of arctic foxes in Fennoscandia (northern Scandinavia and the Kola Peninsula) are vital for the future persistence of the entire Scandinavian population, as they can enhance gene flow by providing a link to Siberia. Previous studies have assessed the differentiation between these populations, but were limited to microsatellites, mitochondrial DNA control regions and mitogenome sequences [39,40,43,44]. It has been established that dispersal with gene flow has gradually declined following the 19th century population bottleneck [39,40,43,48,50]. The state of the arctic fox population on the Kola Peninsula has been unknown for more than half a century, but recent investigations revealed a situation similar to Scandinavia, with an extremely isolated population consisting of at most a few dozen individuals [49]. Likewise, recent assessments of the northernmost subpopulations of Norway in Finnmark suggest that climate-driven rodent cycle irregularity and increasing red fox abundance may have resulted in the near loss of the arctic fox of this area [51]. However, it is likely that the species was still more abundant in Finnmark and on the Kola Peninsula until the 1970s and that the decline was more recent than further south in Scandinavia [49,51].

Reflecting the barriers in Fennoscandia, and resulting population isolation, inbreeding depression was documented in the southern Swedish subpopulation [52]. While limited genetic rescue was detected following immigration from Norway [53,54], research suggests that introducing unscreened genetic variation may impose a trojan horse effect in small populations, where introduced additional deleterious variants risk being unmasked by inbreeding, increasing the mutational load [55,56,57,58]. Therefore, it is vital to consider the proportion of mutations in small, isolated subpopulations relative to larger, outcrossing populations. Inbreeding within the Scandinavian arctic fox populations have mostly focused on the southernmost subpopulation in Sweden. However, connectivity through the north to the large populations in Siberia has been understudied and may be crucial for long-term population persistence and re-establishing a viable metapopulation in Fennoscandia. The genomic consequences of this fragmentation are yet to be explored and could reveal the complexities surrounding its isolation and subsequent decline. The relatively recent fragmentation of the northern subpopulations, which experienced a more recent decline, provides a unique opportunity to explore the levels of genomic inbreeding in small, isolated subpopulations in relation to a much larger, panmictic population. By identifying the levels of genomic inbreeding in northernmost Fennoscandia, the status of these vital subpopulations can be identified. Moreover, relating this information to the outbred Russian population and the highly inbred southernmost Swedish subpopulation will reveal the potential for gene flow and pinpoint high priority areas for conservation measures. Given the small and isolated state of the Fennoscandian subpopulations, the consequences of genetic drift should also be investigated to document the state and accumulation of deleterious mutations.

The aim of this study was to use high-coverage genomes from northernmost Fennoscandia, the neighboring Russian population on Kola and the large populations to the east to assess the genomic consequences of population fragmentation. We determined the extent of genome-wide heterozygosity and inbreeding levels (F_ROH_), as well as the proportion of homozygous versus heterozygous deleterious mutations in the northernmost Scandinavian subpopulation, the Kola Peninsula, and the larger Siberian population. We hypothesized that (1) measures of genomic inbreeding due to common ancestors would be significantly higher in northern Fennoscandia compared to Siberia, (2) being the farthest from Siberia, the northern Swedish population would have the highest levels of recent inbreeding, and (3) the proportion of homozygous deleterious mutations would be significantly higher in northern Fennoscandia, while the proportion of heterozygous deleterious mutations would be significantly higher in Siberia.

## 2. Materials and Methods

### 2.1. Study Populations

The study area spans northern Eurasia from Scandinavia to Eastern Siberia (Figure 1). In northern Sweden, we included samples from Vindelfjällen and Arjeplog, and in northern Norway, we included samples from Saltfjellet, Reisa nord, Varangerhalvøya and Øvre Dividal (Table 1, Figure 2). In Russia, we included one sample from Kola, which is biogeographically part of Fennoscandia, and samples from several sites in Siberia (the region east of the Ural Mountains): Yamal, Taymyr, Indigirka, Faddeyevsky Island and Wrangel Island (Table 1, Figure 1). Samples included in this study were collected between 1989 and 2019 (Table 1), consisting of ear tissue from ear-tagging, preserved in 99% ethanol, and muscle tissue from carcasses, all stored at minus 20 °C. Since the 1980s, the Swedish arctic fox population has been monitored closely, with summer inventories taking place to ear-tag, collect tissue for DNA analysis and monitor reproduction, survival, and body condition of individuals, using unique ear combinations for identification [59]. The Norwegian samples were derived from founders from The Norwegian Arctic fox captive breeding program (Saltfjellet, Reisa Nord and Øvre Dividal), as well as an arctic fox carcass from the Norwegian monitoring program (Varanger Peninsula), implemented by the Norwegian Institute for Nature Research (NINA). 

All Siberian samples were collected between 1994–2008. The Arctic Research Station in Labytnangi contributed two Siberian samples from Yamal (ID 5974, 5967; samples obtained from fur trappers from the indigenous community of Yamal), and the rest were collected during expeditions arranged by the Swedish Polar Research Institute in 1999 and 2005, during the Swedish-Russian Tundra Ecology-Expedition in 1994, and during the International Polar Year in 2008.

### 2.2. DNA Extraction and Resequencing

Complete genome data was collated with existing genomes sequenced within ongoing research (8 Siberian, 1 from the Kola Peninsula, and 22 northern Scandinavian individuals). This data set was supplemented with material from 12 new samples sequenced for this study (4 Siberian and 8 northern Scandinavian individuals), for a total sample size of 43 genomes (Appendix A). Extraction of DNA for the northern Swedish, northern Norwegian and Siberian tissue samples took place using the DNeasy Tissue Kit (Qiagen, Hilden, Germany) according to manufacturer’s instructions in a laboratory separated from post-PCR products. However, 3 Swedish samples were extracted using the Thermo Scientific KingFisher Cell and Tissue DNA kit (Thermo Fisher Scientific, Waltham, USA). After quality check, the Swedish and Siberian genomes were sequenced at The National Genomics Infrastructure (NGI), SciLifeLab, Stockholm, using a combination of the Illumina Novaseq6000 and MiSeq platforms with a 2 × 150 bp setup and TruSeq PCR-free library construction. The 3 Swedish samples extracted using the Thermo Scientific KingFisher Cell and Tissue DNA kit (Thermo Fisher Scientific, Waltham, USA) were sequenced using the Illumina HiSeq X platform with TruSeq library construction, and the library of sample 11119 was constructed using the Lucigen NxSeq AmpFREE Low protocol (Lucigen, Middleton, USA). The Norwegian and 2 Siberian samples (ID 5974, 5967) were sequenced at the Genomics Core Facility, Dept. of Cancer Research and Molecular Medicine at NTNU in Norway. Library preparation for these samples was performed according to the TruSeq DNA PCR-Free LT Sample Preparation Kit (Ilumina, San Diego, CA, USA), and the samples were sequenced on Illumina HiSeq 4000.

### 2.3. Bioinformatics and Variant Calling

For bioinformatic analyses and variant calling, we followed the GenErode pipeline by [62], parameterized according to Hasselgren et al. (2021). First, Trimmomatic v.0.36 [63] was used to trim adapters from raw reads. The trimmed reads were aligned with an arctic fox reference genome (Vlagopus_NRM_v1.fa; Genbank assembly ID: GCA_018835635.1) using the Burrows–Wheeler Aligner (BWA) v.0.7.17 [64]. Samtools v.1.9 [65] was used to sort and index BAM files, duplicates were removed using Picard v.2.10.3 [66] and indels realigned with GATK v.3.7 [67,68]. To control for quality and calculate mean coverage, we used Qualimap v.2.2 [69].

Variants were called using bcftools mpileup v.1.6 and bcftools call v.1.6 [65,70] following a minimum depth of one third of the average coverage (i.e., ~8.89X) and maximum of twice the average coverage for each sample (i.e., ~53.36X). Single Nucleotide Polymorphisms (SNPs) were filtered by base quality equal or higher than QV = 30 and those within 5 bp of indels. The X chromosome, mitogenome, hard masked repeat regions and scaffolds smaller than 25,000 bp long were all removed for downstream analyses using BEDtools v.2.27.1 [71]. All individual vcf files were then merged, obtaining 9,165,115 high quality single nucleotide polymorphisms (SNPs). After merging of vcf files, 13,102,461 high-quality SNPs were obtained. A total of 4,862,746 high quality SNPs were used in downstream analyses, using only variants called in every individual.

### 2.4. Population Structure

Population stratification based on variation in 4,862,746 high quality SNPs was illustrated with a principal component (PC) analysis using PLINK v.1.9, based on the variance-standardized relationship matrix [72], visualized in R v.4.1.2 [73].

### 2.5. Genome-Wide Heterozygosity and Genomic Inbreeding Coefficients (F_ROH_)

The population scaled mutation rate theta (θ = 4 N_e_μ [74]) was estimated using mlrho [75], an approximate measure of expected heterozygosity per site under the infinite sites model, where N_e_ denotes the effective population size and µ denotes mutation rate [76]. Bases with quality below 30, reads with mapping quality below 30, and positions with root-mean-squared mapping quality below 30 were filtered out. Due to structural variation, high and low coverage in certain regions can result in inaccurate mapping to the reference genome and false heterozygous sites; therefore, we filtered out sites with depth one third of the average coverage and twice the average coverage for each sample.

Levels of inbreeding were estimated based on runs of homozygosity (F_ROH_) using a sliding window size of 200 SNPs (0.2 kb; homozyg-window-snp 200) with a maximum of 3 heterozygous SNPs in each window, using the software PLINK v.1.9 [72]. Sliding windows were classified as homozygous if not more than three sites were heterozygous per window (homozyg-window-het 3). A SNP was classified as within a homozygous segment of a chromosome if at least 5% of all windows including that SNP were defined as homozygous (homozyg-window-threshold 0.05). The correct boundaries of a ROH were ensured by using this threshold. Certain conditions determine if a homozygous segment is determined as a ROH: The segment contained ≥100 SNPs long (homozyg-snp 100) and covered ≥100 kb (homozyg-kb 100). Also, it had a minimum SNP density that was one SNP per 50 kb (homozyg-density 50) and the maximum distance between two neighboring SNPs was ≤1000 kb (homozyg-gap 1000).

To determine the number of generations, in time, that inbreeding extends to, we quantified the amount of genomic inbreeding (F_ROH_). This was calculated as the proportion of the total genome that consisted of runs of homozygosity:g = 100/(2rL),(1)

The number of generations is denoted by g, L denotes the length of ROH in Mb, and r denotes the recombination rate (r = 4 N_e_C [77]), where C is the probability of recombination. The recombination rate of the silver fox (*V. vulpes*; 0.6 cM per Mb [78]) was used due to the similar life history traits. Based on this, ROH larger than 100 kb reflects inbreeding that can be traced to ancestors around 850 generations back in time (historical), while 2 Mb reflects around 45 generations, marking the persecution and subsequent endangered status of the Scandinavian arctic fox. ROH larger than 8 Mb reflects coalescence 10 generations back (recent), which corresponds to the time when conservation actions were initiated [61]. Furthermore, due to measuring ROH in Mb instead of genetic map length, (cM), a large variance is expected in the estimates. We used a generation time of 2 years, consistent with Hasselgren et al. [53]. To rule out depth of coverage influencing estimates of F_ROH_ and heterozygosity [53], we tested for relationships between these parameters, with r values varying from −0.06 and 0.29.

### 2.6. Mutational Load

We quantified the mutational load according the Kutschera et al. [62], using the red fox annotation (GCF_003160815.1 [79]) to identify and annotate synonymous and nonsynonymous variants within the proximity of coding regions. Identified mutations were genotyped and categorized into three types by their effect on protein expression: variants unlikely to change protein behavior (low: synonymous), neutral variants that may change protein effectiveness (moderate; missense), and variants that exhibit a high disruptive impact on protein function (high: Loss of Function; LoF). The proportion of variants in each category was calculated by dividing the number of homozygous and heterozygous variants separately in each category by the total number of alleles.

### 2.7. Statistical Analyses

We tested for differences in genome-wide heterozygosity and F_ROH_, along with the proportion of LoF and missense variants between northern Fennoscandia and Siberia, and among each subpopulation using non-parametric Kruskal–Wallis tests, with Dunn’s tests for post-hoc comparisons using the R package rstatix (v.0.7.0 [80]). Tests on F_ROH_ were conducted, first including ROH larger than 100 kb (coalescence < 850 generations back), then including only ROH larger than 2 Mb (<45 generations back) and finally, 8 Mb (<10 generations back).

## 3. Results

### 3.1. Population Structure

We sequenced whole-genomes of 43 arctic foxes at a depth of coverage ranging between 8–38X, with an average of 24X (Appendix A). The principal component (PC) analysis revealed a clear isolation-by-distance pattern in PC1 represented by clustering of the northern Scandinavian samples to the right, with the Vindelfjällen sample from 1989 and the more north-eastern sample from the Varanger Peninsula breaking off from the main cluster (Figure 3a). The Kola sample showed an intermediate position, reflecting the geographic location of Kola in the very east of the Fennoscandian Peninsula (Figure 1). The mainland Siberian samples mainly clustered together, except for one Yamal sample reflecting the geographic distance between Yamal and the other Siberian sample areas further to the East. Wrangel Island clustered with the mainland Siberian samples. Additional structure was revealed by PC2 within the Siberian samples, whereas there was no variation within the northern Scandinavian samples. The sample from Faddeyevsky Island showed substantial separation from the mainland Siberian population. PC3 revealed further structure within the Siberian samples (Figure 3b), congruent with the high levels of genetic variation compared to the northern Fennoscandian samples. Additionally, the Wrangel Island sample separated from the mainland cluster to the top of the PC3 axis.

### 3.2. Genome-Wide Heterozygosity

The genome-wide heterozygosity, calculated as the number of heterozygous SNPs per 1 kb (Appendix A), was estimated to be on average 18.42% higher in the Siberian than northern Fennoscandian samples (Figure 4a), 13.85% higher in the Siberian than in the northern Norwegian samples and 13.81% higher than in the Kola sample when separated (Figure 4b). Northern Sweden had significantly lower genome-wide heterozygosity than Yamal, Taymyr, and E Siberia (χ^2^ = 26.378; *p* = 0.025; *p* = 0.003; *p* = 0.025; Figure 4b, Appendix A). There was, however, no significant difference detected among the remaining sample areas (Appendix A).

### 3.3. Genomic Inbreeding Coefficients (F_ROH_)

On average, northern Fennoscandia had 1.47-fold longer runs of homozygosity (ROH) than Siberia (Figure 5a). Average ROH length declined eastwards, marking a 24.19% reduction from northern Sweden to northern Norway, 38.33% reduction from northern Scandinavia to Kola and a 33.92% reduction from Kola to E Siberia (Figure 5b). Very long ROH (>62 Mb) were detected in ten northern Scandinavian individuals, spanning across almost the entire scaffold. The longest ROH (104.7 Mb) was found in an individual from Arjeplog in northern Sweden, sampled in 2019. The longest ROH amongst the northern Norwegian samples (75.2 Mb) was found in an individual in Dividalen from 2005. Within Siberia, the longest ROH ranged from 8.8 Mb in Taymyr to 27.7 Mb in Yamal.

Individual genomic inbreeding coefficients (F_ROH_), estimated as the proportion of the genome consisting of ROH (Appendix A), ranged from 0.02 to 0.38 for inbreeding traced to both historical and recent ancestors (<850 generations, ROH > 100 kb), and from 0.00 to 0.25 for inbreeding from recent ancestors only (<10 generations, ROH > 8 Mb). There was a relatively high proportion of inbreeding traced to historical ancestors in all the sample areas, increasing towards northern Sweden (Figure 5b). In fact, inbreeding due to historical ancestors was almost as high as recent inbreeding in Fennoscandia. This contrasts with the relatively low rate of inbreeding at 10–45 generations (ROH = 2–8 Mb), which was expected to be highest given the population bottleneck. In northern Fennoscandian arctic foxes, 40% of the individual inbreeding was due to close relatedness between ancestors less than 10 generations back, whereas Siberian foxes only had 7.7% inbreeding due to recent common ancestors (Figure 5a). On average, Siberian individuals had 97.18% lower F_ROH_ than Fennoscandian foxes considering inbreeding from recent ancestors (<10 generations, ROH >8 Mb; χ^2^ = 25.526, *p* ≤ 0.001; Appendix A), 94.34% lower considering inbreeding from up to 45 generations back (>2 Mb; χ^2^ = 25.364, *p* ≤ 0.001; Appendix A), and 85.34% lower considering inbreeding from historical ancestors (<850 generations, ROH >100 kb; χ^2^ = 25.364, *p* ≤ 0.001 Appendix A).

### 3.4. Mutational Load

Northern Fennoscandia exhibited a significantly greater proportion of homozygous LoF and missense mutations than Siberia (χ^2^ = 12.581; *p* = <0.001; Figure 6a; χ^2^ = 25.364; *p* = <0.001; Figure 6c). Moreover, there was a significantly higher proportion of heterozygous LoF and missense mutations in Siberia than in northern Fennoscandia (χ^2^ = 19.96; *p* = <0.001; Figure 6b; χ^2^ = 23.229; *p* = <0.001; Figure 6d). Furthermore, northern Sweden had a significantly higher proportion of homozygous LoF variants than northern Norway and Taymyr (χ^2^ = 25.864; *p* = 0.02; *p* = 0.001; Appendix A), while there was a significantly higher proportion of homozygous missense variants in northern Sweden than in Yamal, Taymyr, and E Siberia (χ^2^ = 26.988; *p* = 0.03; *p* = 0.003; *p* = 0.04; Appendix A). In contrast, northern Sweden had a significantly lower proportion of heterozygous LoF variants (χ^2^ = 22.41; *p* = 0.02; Appendix A), and missense variants (χ^2^ = 24.54; *p* = 0.01; Appendix A) than Taymyr. Northern Norway, however, only had a significantly lower proportion for heterozygous LoF variants than Taymyr (χ^2^ = 22.41; *p* = 0.02; Appendix A).

## 4. Discussion

The primary aim of this study was to assess the genomic consequences of fragmentation of the northernmost Fennoscandian arctic fox populations in relation to the large, likely well-connected Siberian populations. Our study highlights the power of estimating genomic inbreeding and provides a unique historical perspective of demographic processes and genomic consequences of inbreeding up to 850 generations back in a population exposed to both geographic and demographic fragmentation.

Consistent with theoretical predictions and the population history of the arctic fox, there was a significant increase in inbreeding due to recent common ancestors between northern Fennoscandia and Siberia, with a gradual decrease in genome-wide inbreeding coefficients (F_ROH_ > 100 kb) from northern Sweden to eastern Siberia (Figure 5). The average ROH was ~1.47-fold longer in northern Fennoscandia and extensive inbreeding was detected in numerous northern Scandinavian individuals, spanning almost entire scaffolds. In northern Sweden, the extent of inbreeding due to very recent common ancestors (104.7 Mb) was comparable to the levels found in a highly inbred gray wolf (*Canis lupus*) population with ROH reaching 95.8 Mb, where entire chromosomes were also homozygous [81].

Genome-wide heterozygosity was consistent with F_ROH_ estimates, revealing significantly lower levels in northern Sweden compared to the Siberian samples (Figure 4). However, the northern Norwegian and Kola samples had intermediate values and did not differ significantly from any of the other samples. Where F_ROH_ provides a clear estimate of inbreeding due to common ancestors, genome-wide heterozygosity is influenced by a population’s entire demographic history, such as bottlenecks, genetic drift, linkage, dispersal with gene flow and selection [82], as well as life history strategies [83]. Nonetheless, a clear pattern consistent with a stepping-stone model of population structure is observed [84], where decreasing heterozygosity westwards through the Fennoscandian Peninsula implies increased population fragmentation and reduced gene flow. This pattern was consistent throughout the entire history of inbreeding inferred by F_ROH_, becoming more extreme with inbreeding due to recent common ancestors (>8 Mb).

Interestingly, there was a gradual drop in inbreeding due to recent common ancestors eastwards, where average ROH was 38.25% shorter in Kola than in northern Scandinavia. Although, at present, the situation of the Arctic fox on the Kola Peninsula is probably similar to that of northern Scandinavia, it is likely that the population decline happened later than further west, which would explain a lower level of inbreeding. In addition, the sample from the Kola Peninsula dates from the 1990s and, thus, does not reflect the inbreeding in the present-day population. Average ROH were 33.92% shorter in eastern Siberia than in the Kola individual; however, it is important to note that ROH in the Siberian samples are based on limited sample sizes, therefore, results may not be representative.

Nevertheless, these results strongly suggest that subpopulations within northern Fennoscandia are becoming increasingly fragmented and isolated over time. Gene flow from Siberia over the western part of Russia into Kola and northern Scandinavia is thus limited, effectively cutting Scandinavia off from the historically circumpolar distribution. Dalén et al. [43] found low levels of dispersal between Scandinavia, the Kola Peninsula and northwestern Russia, suggested to be driven by recent fragmentation. Moreover, Nyström et al. [39] observed a clear reduction in genetic variation in Scandinavia following the severe 19th century bottleneck. Consequences of fragmentation were evident as genetic differentiation doubled between Scandinavia and northern Russia. Since the time of these studies, the northern Scandinavian subpopulations may have deteriorated even further, with increased isolation and inbreeding over the past 16 years.

Along with the overall pattern described here, we observed a historical trend where inbreeding, due to common ancestors 45–850 generations back, increased southward and westwards towards Scandinavia (Figure 5). Relative to the low levels of inbreeding due to recent common ancestors (<45 generations), Siberian samples had a high proportion of inbreeding due to common ancestors 45–850 generations back (Figure 5a). A similar pattern was seen in northern Norway, where a higher proportion of inbreeding dated back to common ancestors in the deep past (45–850 generations), compared to recent years (<10 generations; Figure 5b). This was unexpected, as most inbreeding due to common ancestors was predicted to trace back to the past 45 generations in northern Scandinavia, corresponding to the population bottleneck of the late 19th century. This may suggest some recent gene flow into these subpopulations, erasing the signature of recent inbreeding (<10 generations). In contrast, in northern Sweden, levels of inbreeding due to common ancestors dating back to 10 generations back were slightly higher than for inbreeding from 45–850 generations back. However, the proportion of inbreeding due to common ancestors tracing back to 10–45 generations was much less apparent (Figure 5b). Although speculative, this unexpected considerable inbreeding in the deep past may correspond to a yet unidentified population bottleneck predating the persecution in the 19th century in the northern Fennoscandian and Siberian populations dating back some 45–850 generations ago. Scandinavia has undergone several warming periods over the past 4000 years, most recently for 200 years between 1200–1400 AD [85], potentially coinciding with this inbreeding event.

A clear differentiation was observed between the northern Scandinavian and Siberian genomes (Figure 3), with greater genetic variation evident within Siberia, an expected characteristic of an outbred and panmictic meta-population. Interestingly, the oldest Vindelfjällen sample sequenced from 1989 clustered with the Varanger sample, which may be a signature of higher connectivity in the past. Moreover, the PCA illustrated that even though the samples were collected over a relatively large time frame (i.e., some range), there is limited temporal genetic change, which is apparent from the clustering of Scandinavian samples. Further, the strong divergence from Siberia adds evidence that little to no gene flow has occurred from east of the Kola Peninsula in recent years. The rather large geographical gap in our sampling, lacking arctic foxes from the western range of the Russian Arctic, may have contributed to this apparent divergence.

Another consequence of isolation at low population size is the accumulation of deleterious mutations. Hasselgren et al. [86] highlighted an important role of deleterious mutations influencing fitness in the southernmost subpopulation in Sweden. Here, we found that northern Fennoscandia harbored significantly more homozygous deleterious mutations, while Siberia had more deleterious variants in heterozygous state, for both moderately and strongly deleterious mutations. These results are consistent with the bottleneck history and estimated inbreeding, leading to deleterious variation becoming increasingly expressed as recessive variants are unmasked. Moreover, it has been discussed in several recent studies that large populations will have a high mutational load; however, the high proportion of recessive deleterious alleles will be masked in heterozygous state [87,88], which has been confirmed by numerous empirical studies [55,56,57,58]. Our results are thus consistent with these findings and further highlight the importance of considering measures of inbreeding and mutational load in small and isolated Fennoscandian arctic fox populations.

Our results provide compelling evidence that population fragmentation leads to genome erosion. In order to understand this issue in more depth, a next step should explore to which extent ROH contribute to the expression of deleterious mutations and to what degree inbreeding depression affects the fitness of the northern Fennoscandian arctic fox populations. Candidate mutations that contribute to inbreeding depression within ROH should be identified [86,89], and the efficiency of purging investigated [22]. Furthermore, the distribution of ROH could allow us to infer the timing of fragmentation in these arctic fox populations, that lack long-term monitoring data, particularly on the Kola Peninsula [49,90]. A temporal approach [91] may also illustrate the genetic response to historical events [57,58,92,93].

## 5. Conclusions

Our study highlights the genomic consequences of isolation and long-term low population size in the arctic fox populations of northern Fennoscandia, suggesting more severe population decline and genome erosion than identified in previous studies. While there was a past connection between northern Scandinavian foxes over the Kola Peninsula and the White Sea to the large populations in the Russian Arctic, [40,48], gene flow into Scandinavia via the Kola Peninsula appears to have been drastically reduced in the recent past (i.e., ~20 years). In addition to the very small population sizes, the dramatic decrease in sea ice on the White Sea in the first part of the last century may have contributed to this geographical isolation [49]. The arctic fox subpopulations on the Varanger Peninsula and Kola Peninsulas are exceptionally isolated and in low abundance [49,51]. In situ conservation measures such as supplemental feeding should be prioritized in a stepping-stone fashion to restore connectivity and promote natural dispersal [44], along with ex situ measures such as restocking high priority areas with healthy individuals from nearby areas or with captive-bred individuals to increase heterozygosity and reduce the frequency of detrimental variation. Such intensive measures are, since 2018, implemented on the Varanger Peninsula. However, functional genomic regions should first be screened for population-specific deleterious mutations to determine the likelihood of genetic rescue via heterosis versus the risk for introducing additional deleterious variants, leading to potential unintended fitness consequences (e.g., [86]).

Fragmentation of Arctic populations can be strengthened by invasion of boreal species. Following warming in the Arctic, and intensified human activities, the red fox has encroached into the low-Arctic tundra of Vindelfjällen, the Varanger Peninsula and some places in Siberia [94,95,96]. These areas are vital for maintaining connectivity between Scandinavia and the Russian Arctic, and with high levels of inbreeding detected, this invasion may be inhibiting dispersal of arctic foxes among subpopulations [42]. To reduce this risk, red fox culling could be implemented in high priority areas in the western range of the Russian Arctic [49]. Arctic foxes may be unable to track climate-induced habitat change [97], thus, isolated refugia is a likely future consequence of a warming Arctic.

Given the extent of historical and recent inbreeding, the northern Fennoscandian population is likely to continue to genetically deteriorate and decline. The arctic fox has a history of low genetic variation [98,99], hypothesized to result from repeated isolation in refugia during interglacial periods or local extinctions during range contraction [40,47]. Based on our results, this low genetic variation could also be a consequence of historical demographic bottlenecks following the last glacial maximum and preceding the late 19th century bottleneck. To disentangle the drivers of past population decline and the historical events shaping present day genetic variation, effective population size (N_e_) models should be built to investigate demographic processes over the past 2000 years. Moreover, as the abundance of food resource has a considerable influence on arctic fox demography, it would be particularly relevant to explore how the demographic history of the Norwegian lemming interplays with climatic variation and history of arctic foxes in Fennoscandia. As whole-genome sequencing becomes cheaper and more accessible, complex models of population history may provide answers to the unpredictable population oscillations in Arctic regions. Arctic species must withstand extensive climatic variation and resource fluctuations [100]. With accelerated climate warming, habitat fragmentation and shifting ecosystem dynamics, understanding how species respond ecologically and genetically to these challenges is thus crucial for conservation management.

## Figures and Tables

**Figure 1 genes-13-02124-f001:**
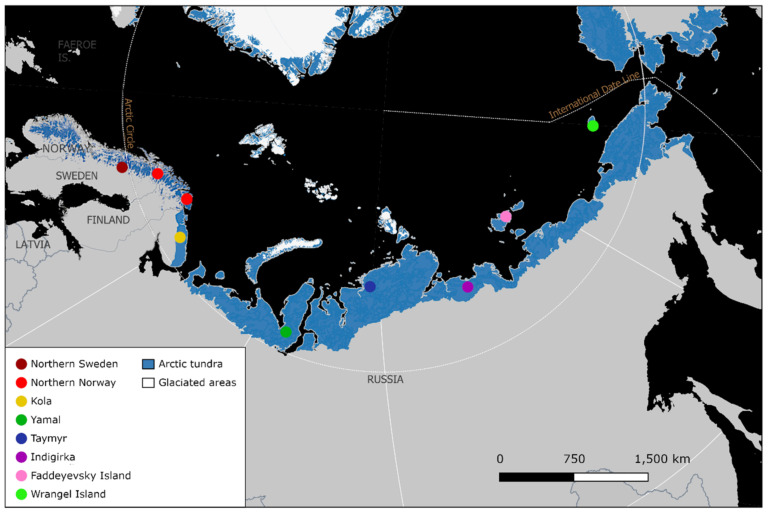
Map showing the distribution of Arctic tundra habitat (Scandinavia: alpine tundra [60]; Eurasia: oro-arctic tundra [61]) in relation to study sample areas in northern Sweden, northern Norway, Kola, Yamal, Taymyr, Indigirka, Faddeyevsky Island and Wrangel Island.

**Figure 2 genes-13-02124-f002:**
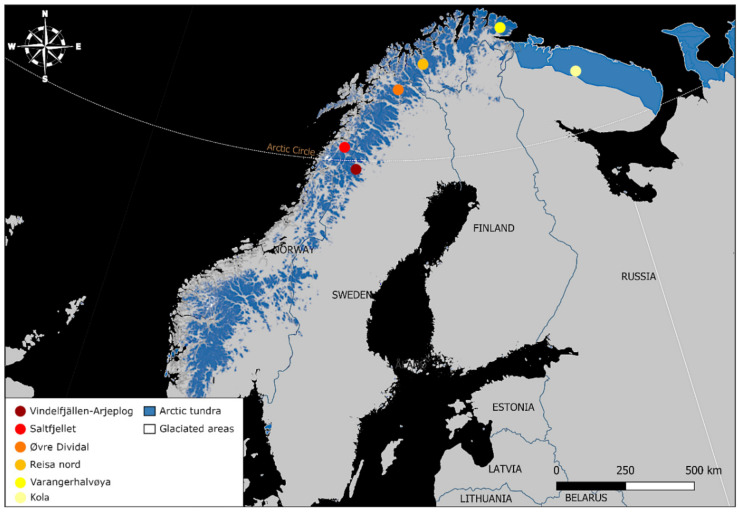
Map showing the distribution of Arctic tundra habitat (Scandinavia: alpine tundra [60]; Kola: oro-arctic tundra [61]) in relation to study sample areas in Vindelfjällen-Arjeplog, Saltfjellet, Øvre Dividal, Reisa nord, the Varanger Peninsula, and the Kola Peninsula.

**Figure 3 genes-13-02124-f003:**
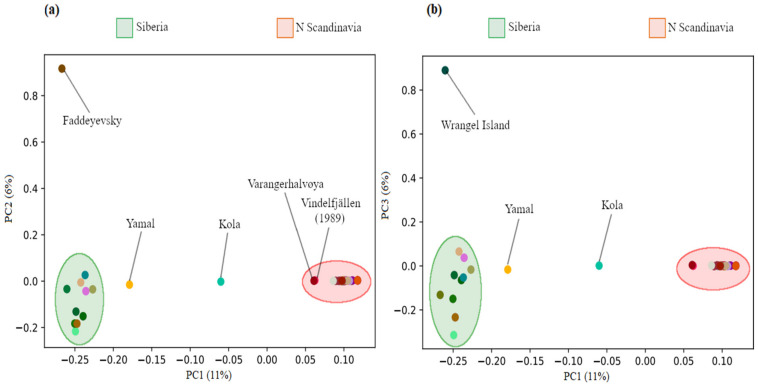
Principal component (PC) analysis of population stratification across samples of arctic foxes on axes (**a**) PC1-PC2 and (**b**) PC1-PC3. Points represent genotypic data for 4,862,746 single nucleotide polymorphisms (SNPs) per individual. The first two principal components (PCs) explained 11% (PC1) and 6% (PC2) of the genotypic variation across all individuals and SNPs. A clear isolation-by-distance pattern is exhibited. PC3 explained 6% of the genotypic variation.

**Figure 4 genes-13-02124-f004:**
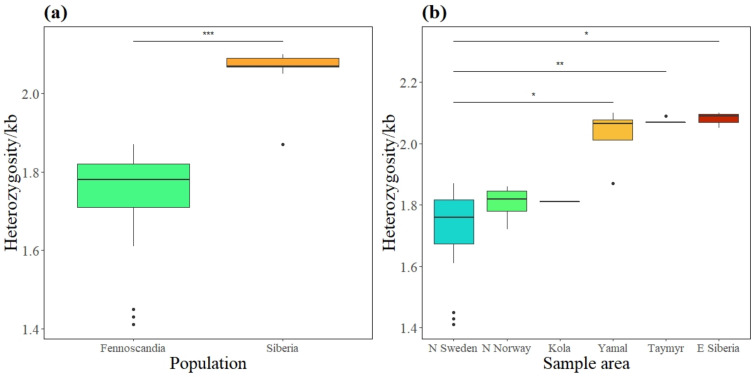
Genome-wide heterozygosity of individuals in northern Fennoscandian and Siberian populations (**a**) and separated into subpopulations (**b**). Heterozygosity measured as the number of heterozygote sites per 1 kb. The horizontal bar shows the median, the boxes show the 25–75% interquartile range and the whiskers show the whole range. Significant results marked with * P < 0.05, ** P < 0.01, *** P < 0.001.

**Figure 5 genes-13-02124-f005:**
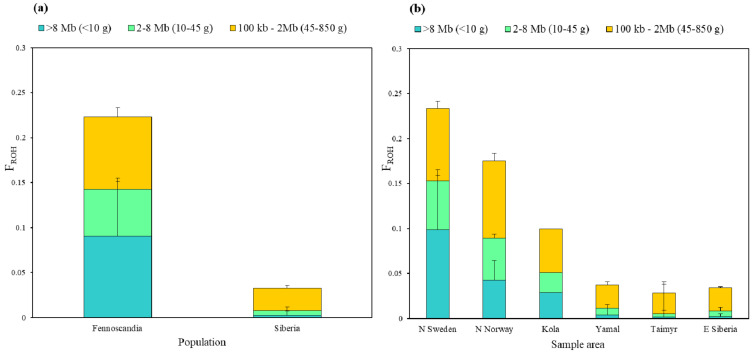
Mean genomic inbreeding coefficients (F_ROH_) with standard deviation for individuals in northern Fennoscandian and Siberian populations (**a**) and separated into subpopulations (**b**). Orange bars show inbreeding due to close relations deep back in history (45–850 generations, >100 kb–2 Mb), Green bars shows show inbreeding due to close relations 10–45 generations back (2–8 Mb) and blue bars show inbreeding due to recent ancestors less than 10 generations back (>8 Mb).

**Figure 6 genes-13-02124-f006:**
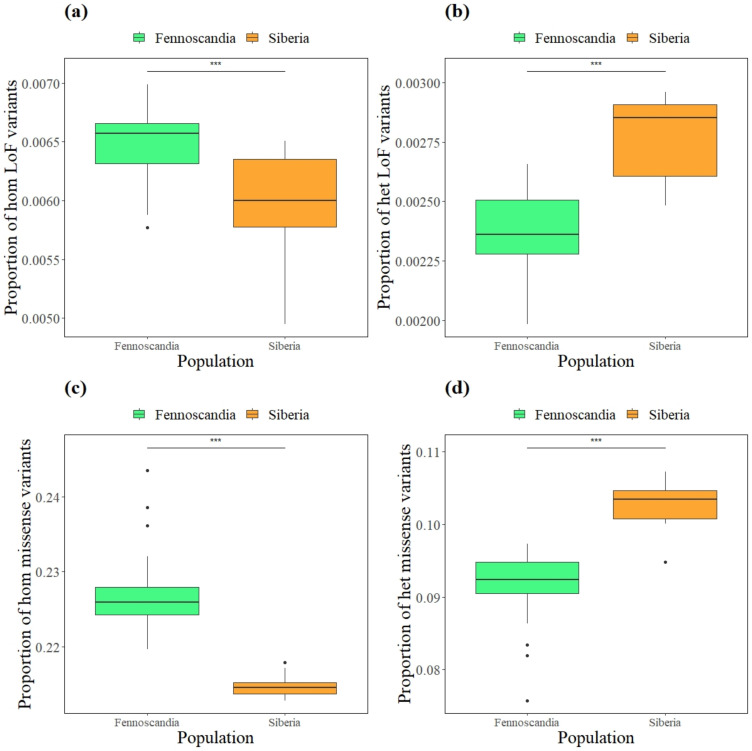
Proportion of deleterious variants in arctic foxes of the northern Fennoscandian and Siberian populations; separated into (**a**) homozygous loss of function variants (LoF), (**b**) heterozygous LoF variants, (**c**) homozygous missense variants, and (**d**) heterozygous missense variants. Proportion calculated as the number of each variant type divided by total variants. The horizontal bar shows the median, the boxes show the 25–75% interquartile range and the whiskers show the whole range. Significant results marked with *** P < 0.001.

**Table 1 genes-13-02124-t001:** Summary of country, area, year, tissue type, sampling method and number of individuals used for whole genome analyses, with new samples sequenced for this study in parentheses.

Population	Country	Sample Area	Year	Tissue Type	Sampling Method	*n*
Fennoscandia	Northern Sweden	Vindelfjällen	1989–2019	Ear tissue	Ear tagging	15
		Arjeplog	2008–2019	Ear tissue	Ear tagging	11(5)
	Northern Norway	Saltfjellet	2007	Ear tissue	Ear tagging	1
		Reisa nord	2007	Ear tissue	Ear tagging	1
		Varangerhalvøya	2011	Muscle tissue	Injured fox	1
		Øvre Dividal	2005	Ear tissue	Ear tagging	1
	Russia	Kola	1990s	Muscle tissue	Carcass sampling	1
Siberia	Russia	Indigirka	1994	Muscle tissue	Carcass sampling	1(1)
		Yamal	1994–2011	Skin/muscle tissue	Carcass sampling	4
		Taymyr	1994	Skin/muscle tissue	Carcass sampling	5(4)
		Faddeyevsky Island	1994	Muscle tissue	Carcass sampling	1(1)
		Wrangel Island	2008	Skin tissue	Carcass sampling	1(1)

## Data Availability

The whole-genome sequencing data are available to download at the European Nucleotide Archive (ENA) under accession number PRJEB55788.

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
