# Peer review of "Genomic Consequences of Fragmentation in the Endangered Fennoscandian Arctic Fox (Vulpes lagopus)"

_genes, 2022, doi:10.3390/genes13112124_

Round 1
Reviewer 1 Report
The manuscript by Cockerill et al. reports on a study of whole genome data from arctic foxes across their current range. Genome data was already available from 31 individuals and was combined with whole genomes generated from another 12 samples collected as part of this study to test hypotheses about genome erosion associated with isolation by distance and population fragmentation. Important samples from northern Fennoscandia linking Scandinavian populations with those in Siberia were included to provide insights about current geneflow and the accumulation of deleterious mutations.
The study highlights large differences in heterozygosity between Siberian samples and those from Fennoscandia. Inbreeding as estimated from runs of homozygosity also indicate large regional differences, with Scandinavian samples and those from the Kola Peninsula showing strong signals of inbreeding within the last 10 generations. Interestingly, for Siberian samples and those from northern Norway, there was also an unexpected strong signal of inbreeding in the more distant past, up to 850 generations before present.
The accumulation of deleterious mutations also underlined the consequences of population fragmentation and decline in Fennoscandia. Northern Fennoscandian samples harboured more homozygous deleterious mutations than Siberian samples where deleterious variants appeared at a higher level in the heterozygous state. Overall, the study highlights the severe levels of genomic erosion in northern Scandinavia and the need for urgent conservation measures to restore demographic and genomic stability in western portion of the range. The study shows the benefit of whole genome data for estimating inbreeding and informing historical demographic processes. The manuscript is well written and clearly describes a comprehensive and interesting study on the consequences of climate-induced population fragmentation and range reduction. I only have some minor points for improvement.
Sometimes the wording creates confusion within the text, e.g.:
Lines 74-75: “due to the need of in-depth monitoring, unpredictable natural conditions and difficulty obtaining individual fitness data that are linked to genetic variation”. Consider changing to “due to the need for in-depth monitoring combined with unpredictable natural conditions, and the difficulty in obtaining individual fitness data that are linked to genetic variation”.
Line 82: change “and lacking a historical perspective of inbreeding” to “and limited historical perspectives about inbreeding”
Line 111: replace “what remains are several isolated, genetically distinct subpopulations” with “the few remaining Fennoscandia sub-populations are isolated, genetically distinct units”
Line 116: Provide a reference for the dispersal ability of arctic foxes
Line 160: change “as the proportion and state of deleterious mutations accumulated” to document the state and accumulation of deleterious mutations”
Table 1: Please ad an indication of which samples are the new ones sequenced for this study
Line 300: This is the first use of “LoF”, please give the full term “loss of function” at this point
Figures 4 and 6: I would remove the legends giving the sample locations. This information is provided on the x-axis and the legend just distracts from the figures
Lines 360 to 371: I think it is important here to highlight the high rate of inbreeding attributed to around 850 generations as opposed to that for around 45 generations. As it reads now, the surprisingly low proportion for the 45 generation scale is not really mentioned in the results. Given that you expand on this quite extensively in the discussion, it’s important to draw the reader’s attention to it in the results.
Reviewer 2 Report
1.The study performed the whole genome resequencing analysis for 43 Arctic fox individulas spaning different regions, which will provide useful information for understanding status of genomic consequences of fragmentation and provide insights for its conservation in future.
2.For introduction section, there are too much content. The authors need simplify and highlight the laster progresses related to genetic diversity or conservation of Arctic fox .
3. Title: why not mention the Arctic fox directly?
4.For matierials and methord section, data processing need to be improved and make it more clear.Eg, data filter.
5. The writing of the manuscript need to be improved substantially.
6.For the reulst section,fig.4 and fig.5 is not clear. The colours of the labels are too similar.
Round 2
Reviewer 2 Report
The manuscirpt has been improved largely . But the writing needs to be improved before publication.